# Influence of Skin Commensals on Therapeutic Outcomes of Surgically Debrided Diabetic Foot Infections—A Large Retrospective Comparative Study

**DOI:** 10.3390/antibiotics12020316

**Published:** 2023-02-03

**Authors:** Ilker Uçkay, Dan Lebowitz, Benjamin Kressmann, Benjamin A. Lipsky, Karim Gariani

**Affiliations:** 1Infectiology, Balgrist University Hospital, Forchstrasse 340, 8008 Zurich, Switzerland; 2Service of Infectious Diseases, Geneva University Hospitals and Faculty of Medicine, University of Geneva, 1211 Geneva, Switzerland; 3Service of Internal Medicine, Clinique des Grangettes, Hirslanden, 1224 Geneva, Switzerland; 4Department of Medicine, University of Washington, Seattle, WA 98195, USA; 5Division of Endocrinology, Diabetes, Nutrition and Therapeutic Patient Education, Geneva University Hospitals, 1205 Geneva, Switzerland

**Keywords:** antibiotic therapy, diabetic foot infections, non-beta-lactam antibiotics, skin commensals, treatment failures, associations with treatment failure

## Abstract

In diabetic foot infections (DFI), the clinical virulence of skin commensals are generally presumed to be low. In this single-center study, we divided the wound isolates into two groups: skin commensals (coagulase-negative staphylococci, micrococci, corynebacteria, cutibacteria) and pathogenic pathogens, and followed the patients for ≥ 6 months. In this retrospective study among 1018 DFI episodes (392 [39%] with osteomyelitis), we identified skin commensals as the sole culture isolates (without accompanying pathogenic pathogens) in 54 cases (5%). After treatment (antibiotic therapy [median of 20 days], hyperbaric oxygen in 98 cases [10%]), 251 episodes (25%) were clinical failures. Group comparisons between those growing only skin commensals and controls found no difference in clinical failure (17% vs. 24 %, *p* = 0.23) or microbiological recurrence (11% vs. 17 %, *p* = 0.23). The skin commensals were mostly treated with non-beta-lactam oral antibiotics. In multivariate logistic regression analysis, the isolation of only skin commensals was not associated with failure (odds ratio 0.4, 95% confidence interval 0.1–3.8). Clinicians might wish to consider these isolates as potential pathogens when selecting a targeted antibiotic regimen, which may also be based on oral non-beta-lactam antibiotic agents effective against the corresponding skin pathogens.

## 1. Introduction

Diabetic foot infections (DFI), including osteomyelitis (DFO), are associated with high rates of treatment failure, even when treated with prolonged antibiotic therapy, adequate surgical debridement, and appropriate wound care [1]. There are multiple reasons for the poor outcomes, including limb ischemia, inadequate pressure off-loading of the foot, and a lack of patient adherence to the prescribed treatment [1]. In contrast to what many clinicians believe, the specific causative DFI pathogen is generally not a major determinant for the outcome of therapy [2,3,4,5,6,7,8], unless it is resistant to multiple antibiotic agents [4]. Indeed, in almost all published reports regarding DFIs, clinical or microbiological outcomes are no worse for patients infected with “difficult” pathogens such as methicillin-resistant *Staphylococcus aureus* [2], *Pseudomonas aeruginosa* [3,5], or obligately anaerobic bacteria than with other pathogens [6,7,8]. Even in randomized, controlled trials of treatment of DFI, the causative pathogen(s) is a negligible factor in treatment failure, compared to other parameters [9,10].

Unlike pathogenic bacteria such as *S. aureus*, *P. aeruginosa*, *Enterobacteriaceae*, enterococci, or streptococci, skin commensals isolated from swab cultures are not usually considered true pathogens, even when grown repeatedly from specimens [11,12,13,14]. According to widespread clinical experience and very few retrospective studies, skin commensals [15] (mostly coagulase-negative staphylococci [16,17,18], micrococci [15,16], corynebacteria [15,16,19], and cutibacteria [15,16,20,21]) demonstrate lower clinical virulence than other bacterial genera upon the manifestation of infection. However, there are few published data to inform whether skin commensals are clinically associated with a better outcome after therapy for DFI. We investigate this gap in the literature. 

## 2. Results

### 2.1. Study Population and Infections

All the authors of this paper worked at Geneva University Hospitals during the generation of the scientific database of the Clinical Pathway for DFI. Using that pathway database, we identified 1,018 DFI episodes (median age 81 years, 73% males, 610 [60%] with peripheral arterial disease). Among these, skin commensals were the sole isolates from wound cultures (without any pathogenic pathogens detected) in 54 cases (5%), and in of 23 of these 1018 cases (2% of all cases) the patient was diagnosed as having DFO. The proportion of DFI episodes caused entirely by pathogenic pathogens was 63% (641/1,018). Among these patients whose cultures grew at least one pathogenic pathogen (the control group), the most common isolates were *Staphylococcus aureus* (389 cases [38%]) and *Pseudomonas aeruginosa* (61, 6%), but the cultures yielded 30 other pathogenic pathogens (e.g., β-hemolytic streptococci or *Enterobacteriaceae*). 

Overall, we detected 68 different microbiological constellations. The five most frequent monobacterial, predominant, and pathogenic species were *Staphylococcus aureus* (38%), streptococci (6%), enterococci (5%), and Gram-negative microorganisms (18%), of which *Pseudomonas aeruginosa* was 6%. Skin commensals were retrieved as (co)-pathogens, together with pathogenic bacteria, in 161 DFI cases (16%). Blood cultures grew organisms that we believed represented clinically plausible bacteremia in 80 episodes (8%). The median serum C-reactive protein (CRP) level among all enrolled subjects on admission was 81 mg/L. Among the 392 [39%] episodes of DFO with a positive bone culture, the diagnosis of chronic osteitis was confirmed by histology in 275 cases (70%), while the rest were confirmed by clinical and imaging findings.

### 2.2. Therapy and Outcomes

After treatment (including at least one surgical debridement in all, and partial amputation in 596 [58%], antibiotic therapy [45 different regimens (Figure 1) with a median duration of 20 days, of which 5 days were administered parenterally], hyperbaric oxygen therapy (98 cases [10%]), 251 (25%) of the episodes met our definition of clinical failure. Of these, 119 cases (12%) met our definition of microbiological recurrence. The follow-up duration for these episodes was a minimum of six months, and a median of 3.3 years. The six main antibiotic agents used for skin commensals were co-amoxiclav (40%; practically for all susceptible commensals), vancomycin (15%), co-trimoxazole (10%), clindamycin (8%), doxycyclin (8%), and fusidic acid with rifampicin (8%). Hence, 34% were treated with co-trimoxazole, doxycycline, or rifampicin plus fusidic acid because that was the best choice based on the antibiogram. Linezolid and daptomycin were very rarely used, and only for a short period.

For further analyses, we compared the 54 DFIs solely caused by skin commensals to the 641 DFIs caused solely by pathogenic pathogens (Table 1). As noted, we censored episodes with a mixture between both types of isolates [16] and found no difference in the incidence of clinical failure between the skin commensals and pathogenic pathogens (17% vs. 24%, respectively; *p* = 0.23) or microbiological recurrence (11% vs. 17 %, respectively; *p* = 0.23). Clinically, the study groups only significantly differed in the CRP values at admission (median of 25 mg/L vs. 105 mg/L, respectively; *p* < 0.01). The number of surgical debridement, proportion of DFO cases, occurrence of bacteremia, and the duration of antibiotic therapy (including the parenteral part) were not significantly different between the groups. With further stratifications based on soft tissue DFI and DFO, we found no significant differences in both strata. Treating only the skin pathogens among the cases with only soft tissue infections revealed a similar clinical failure rate as for the pathogenic pathogens (7/24; 29% vs. 94/296; 32%, *p* = 0.85). The same was true for the clinical failure rate for cases with DFO (2/21 vs. 59/192; *p* = 0.10). 

Using multivariate adjustment with the outcome “clinical failure” (Table 2), growth of skin commensals on wound cultures was not a determinant of clinical failure (odds ratio 0.4, 95% confidence interval 0.1–3.8), but the presence of ipsilateral lower extremity ischemia was (OR 3.0, 95% CI 1.1–8.5). These findings were similar in a multivariate analysis for “microbiological recurrence” (Table 2). The Receiver Operating Curve (ROC) value was 0.83, representing a good accuracy of our multivariate model.

## 3. Discussion

In this single-center study, we did not detect any association of the clinical or microbiological outcomes of DFI with the presence of skin commensals compared with pyrogenic pathogens. Furthermore, we found that the number of surgical debridement, the incidence of bacteremia, the percentage of patients with DFO, and the length of antibiotic therapy were quite similar for the two microbiological groups. These results suggest there was not a major comparison bias in management related to the two microbiological groups. The clinical “virulence” of both pathogen groups was similar. Hence, if both patient groups with skin commensals and “pathogens” are equally treated with sensitive antibiotics based on respective antibiograms, there would be no clinical differences.

The only two differences of note between the groups were a significantly lower C-reactive protein level at admission in those with skin commensals and the association of lower extremity ischemia with a higher rate of clinical failure [22] but not microbiological recurrence [23]. Only the choice of the antibiotic agent was different. Secondly, contrary to pathogenic DFI pathogens, for which (oral) co-amoxiclav is the hallmark in the Swiss medical culture [24], we mostly used non-beta-lactam and non-quinolone antibiotic agents that have a similar clinical efficacy as oral beta-lactam agents. We conclude that while skin commensals may induce a lesser degree of inflammation (CRP elevation), they do not appear to be less virulent than the classical bacteria in patients treated for DFI. Thus, there does not appear to be a reason to select less aggressive surgical or antibiotic therapies for DFIs caused by these bacteria.

Besides its retrospective nature, a relatively small sample size of only 60 DFI episodes in the skin commensals’ group, and the large case-mix inherent to the adult DFI population, our study has several other limitations. First, we somewhat arbitrarily created two microbiological groups, one with only skin commensals and the other with only pathogenic pathogens, while in reality two-thirds of skin commensals are co-pathogens with other pathogenic bacteria. However, for formal comparative statistics, we had to exclude mixed-group cases in order to perform a true statistical comparison of sharply distinguished groups of interest. Similarly, our skin commensal classification was composed of many species (e.g., micrococci, *S. epidermidis* [16,18], and *S. lugdunensis* [17])*,* each of which might have a different level of clinical virulence or ability to cause persistent infection. Even with our large number of DFI episodes, it is not impossible to adjust for the effect of a single species in the frequently polymicrobial infections in our study population [16]. 

Second, our analysis may lack other important variables such as the ulcer or infection healing time. Ulcer healing is heavily influenced by off-loading, patient’s compliance, and professional wound debridement, and probably only to a minor extent by antibiotic therapy. The role of pathogens in ulcer colonization, diabetic foot microbiome, and ulcer healing is a matter of debate in human ulcers. The role of the microbial bioburden in ulcer outcomes and complications remains ambiguous, including the significance of microbial load and diversity and the role of specific microorganisms, including known wound pathogens and microorganisms considered as skin commensals or environmental contaminants. In experimental studies, the cultured wound isolates of *S. aureus* elicited differential phenotypes in mouse models that corresponded with patient outcomes, while wound “bystanders” such as *Corynebacterium striatum* and *Alcaligenes faecalis* typically considered commensals or contaminants, also significantly impacted wound severity and healing [25].

Furthermore, as we relied on classical, clinical culture techniques, we might have missed unidentified species within the microbiome [15,26]. These might have been detected by molecular methods such as “shotgun” and other DNA-enhancing techniques [12,15]. There is a growing literature assessing the effects of these “hidden” bacteria (based on standard cultures) within the microbiome or the biofilm. For example, some research groups advocate that these hidden commensals may interact with other bacteria, perhaps even promoting wound healing by inhibiting the virulent *S. aureus* [27] that are so often found in diabetic foot wounds [28,29]. Undertaking such a study would require expensive and limited academic laboratory facilities, making it beyond our routine clinical evaluation.

Lastly, some clinicians might argue that the presence of skin commensals on wound culture is more a sign of specimen contamination than of true infection, or organism selection by prior antibiotic therapy. We do not think this is so, as our diagnostic criteria are based on the IWGDF guidelines [13] and on a high proportion of histologically confirmed DFO episodes. Moreover, on the clinical side, we managed patients with these skin commensals the same as those with every other pathogen, and still saw no difference. If these bacteria play a less virulent role, we think we should have found at least some hints in favor of an altered outcome when studying 1,018 episodes in the same Clinical Pathway.

## 4. Conclusions

In one of the largest single-center case-controlled studies in the field of DFI and DFO, our retrospective results suggest that skin commensals isolated from DFIs or from DFOs, are neither clinically virulent nor more microbiologically persistent than other bacteria. They can also be treated by oral antibiotic agents. Clinicians should therefore perhaps consider these bacteria as potential pathogens when selecting an antibiotic regimen. Similarly, there is probably no need to advocate a different antibiotic treatment (e.g., shorter or longer treatments) when compared to the therapy for pathogenic bacteria. Further clinical confirmatory stories are needed.

## 5. Methods

At the Geneva University Hospitals, we have established a database (embedded in a hospital-wide Clinical Pathway for DFI [1]) for managing DFI. We examined all DFI episodes identified from 24 April 2013 to 31 July 2016 for which microbiological samples were collected. Furthermore, our Clinical Pathway prospectively assessed all DFI and DFO cases that we encountered in the entire hospital. The pathway involved hospitalized patients and those in outpatient settings. All physicians and surgeons were asked to report all DFI patients. Moreover, in the context of the Clinical Pathway implementation, a Research Nurse specialized in DFI regularly screened all hospitalization wards for diabetic patients with and without foot problems, and identified potential DFI candidates. 

We identified all pathogens from these specimens using internationally recommended culture methods [2,3,4]. Wound cultures were only accepted from depth samples (including pus) of the wound after the start of debridement and/or intraoperatively. In the Clinical Pathway conforming to the IWGDF guidance, we avoided superficial microbiological swab sampling [13]. We defined DFIs based on the International Working Group on the Diabetic Foot (IWGDF) criteria [13] and a “clinical failure” as: (1) the persistence or recurrence of any clinical indication for revision surgery; (2) the development of a recurrent infection (same site, same causative pathogen[s]; or (3) the occurrence of a new infection on the same foot [9]. We defined “microbiological recurrence” as a “clinical failure” predominantly caused by the same pathogens as in the index episode. We recorded the three most frequent pathogens per episode, and censored any other quantitatively less common microorganisms. We developed our Clinical Pathway for DFI as a quality program, for which the patients were not required to provide individual consent. However, many of them concomitantly participated in at least one of the many randomized DFI trials we conducted [9,10,30,31] that required signed consent forms.

### Statistical Analyses

For this study, we divided the isolated microorganisms into two groups: those that we regarded, based on the literature and our extensive experience, as only *commensals* (coagulase-negative staphylococci, micrococci, cutibacteria, and corynebacteria); and *pathogenic pathogens* composed of bacteria commonly regarded as virulent and causing DFIs. The primary objective of this study was to define the role of skin commensals in DFIs by examining the likelihood of clinical remission of DFI overall, and diabetic foot osteomyelitis (DFO) separately. We compared the skin commensal and pathogenic pathogen groups using the Pearson-χ^2^ or the Wilcoxon rank sum test, as appropriate. In these comparisons, we only analyzed infections caused entirely by skin commensals and those caused entirely due to pathogenic bacteria, excluding from these group comparisons any polymicrobial DFIs with mixed groups (i.e., pathogenic pathogens AND skin commensals). We furthermore adjusted for our large case-mix with two identical, cluster-controlled (clustering on the individual patient) multivariate logistic regression analyses with the separate outcomes “clinical failure” and “microbiological recurrence”. We performed all statistical calculations using STATA^™^ software (Version 14, College Station, TX, USA). 

## Figures and Tables

**Figure 1 antibiotics-12-00316-f001:**
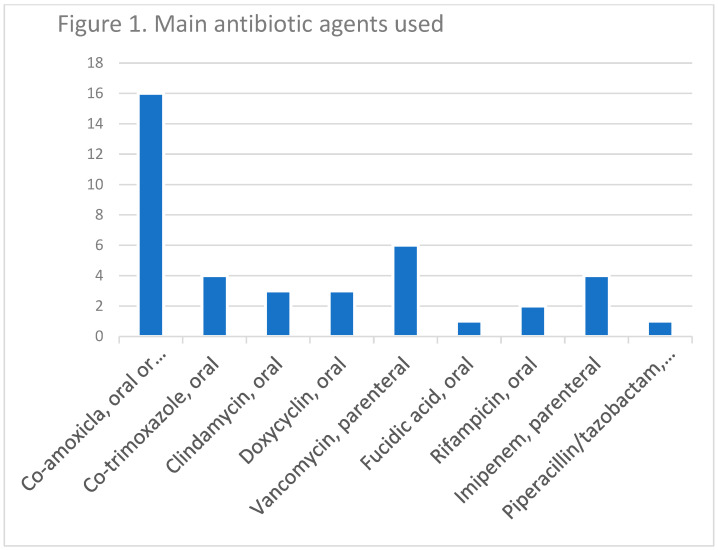
Most frequent antibiotic agents for the treatment of diabetic foot infections.

**Table 1 antibiotics-12-00316-t001:** Comparison of selected factors in patients with diabetic foot infections with *skin commensals* versus *pathogenic bacteria*.

	Pathogenic Bacteria Only		Skin Commensals *^+^*
Factor	n = 641	*p* Value *	n = 54
Median age (years)	80	0.26	83
Osteomyelitis	251 (39%)	0.62	23 (43%)
Bacteremia associated with diabetic foot infection	71 (11%)	0.21	3 (6%)
Median C-reactive protein level on admission	105 mg/L	** *0.01* **	25 mg/L
Median number of surgical debridement	1	0.18	1
Median duration of antibiotic treatment	21 days	0.71	30 days
Median duration of parenteral therapy	6 days	0.88	6 days
Hyperbaric oxygen therapy	73 (11%)	0.19	3 (6%)
Clinical failures (after end of therapy)	153 (24%)	0.22	9 (17%)
Microbiological recurrence (with same pathogens)	111 (17%)	0.24	6 (11%)

*** Significant *p* values ≤ 0.05 (two-tailed) are displayed in bold and italic. ***^+^*** mostly coagulase-negative staphylococci, micrococci, corynebacteria, and cutibacteria.

**Table 2 antibiotics-12-00316-t002:** Logistic regression analyses, stratified based on outcomes of “clinical failure“ and “microbiological recurrence”. Results are expressed as odds ratios with 95% confidence intervals.

Outcome “Clinical Failure”	Univariate	Multivariate	Multivariate	Univariate	“Microbiological Recurrence”
Age	1.0, 1.0–1.0	1.0, 0.9–1.0	1.0, 0.9–1.1	1.0, 1.0–1.0	Age
Number of surgical debridement	0.7, 0.6–0.8	1.2, 0.8–1.8	2.2, 0.7–6.7	1.1, 0.9–1.3	Number of surgical debridement
Total duration of antibiotic therapy	1.0, 1.0–1.0	1.0, 1.0–1.0	1.0, 1.0–1.0	1.0, 1.0–1.0	Total duration of antibiotic therapy
Initial serum C-reactive protein level	1.0, 1.0–1.0	1.0, 1.0–1.0	1.0, 1.0–1.0	1.0, 1.0–1.0	Initial serum C-reactive protein level
Bacteremia	0.6, 0.3–1.1	0.5, 0.1–2.8	1.8, 0.3–3.3	1.4, 0.7–2.6	Bacteremia
Osteomyelitis	0.8, 0.6–1.1	0.8, 0.3–2.1	1.2, 0.3–4.3	0.9, 0.8–1.4	Osteomyelitis
Infection due to skin commensals	0.6, 0.3–1.3	0.4, 0.1–3.8	0.5, 0.1–4.2	0.6, 0.2–1.4	Infection due to skin commensals

## Data Availability

Key data are available in an anonymous form upon reasonable scientific request to the corresponding author. They are not publicly available.

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
