# Peer review of "Influence of Skin Commensals on Therapeutic Outcomes of Surgically Debrided Diabetic Foot Infections—A Large Retrospective Comparative Study"

_antibiotics, 2023, doi:10.3390/antibiotics12020316_

Round 1
Reviewer 1 Report
This is an interesting study trying to understand the role of skin commensals in the outcomes of surgically-debrided diabetic foot infections. I would like to have some explanations and possibly some minor revisions:
1. If the patients and database are from Geneva University Hospitals, which is the contribution to the study of Zurich University Hospital and University of Washington, Seattle? Moreover, where is located the Service of Infectious Diseases: Geneve or Zurich or Seattle?
2. Skin commensals are here contraposed to “pyogenic pathogens”, instead of using the term “pathogenic bacteria” more often used when Enterobacteriaceae and Pseudomonas aeruginosa are included together with Staphylococcus aureus and Streptococci. Please correct line 114 where you can find “pyrogenic” as a mistyping.
3. If both patient groups with skin commensals and pathogens are equally treated with sensitive antibiotics based on respective antibiograms, how can you expect any difference? A targeted antibiotic therapy is the main reason for not finding any difference in the two groups.
4. At the end, an oral non-beta-lactam antibiotic therapy is suggested without giving a clear reason since 40% of the skin commensals group was treated with co-amoxiclav. It can be supposed that 34% were treated with co-trimoxazole or doxycycline or rifampicin plus fusidic acid because that was the best choice based on the antibiogram.
Author Response
Comments and Suggestions for Authors
This is an interesting study trying to understand the role of skin commensals in the outcomes of surgically-debrided diabetic foot infections. I would like to have some explanations and possibly some minor revisions:
Answer: We thank you very much.
- If the patients and database are from Geneva University Hospitals, which is the contribution to the study of Zurich University Hospital and University of Washington, Seattle? Where is located the Service of Infectious Diseases: Geneva or Zurich or Seattle?
Answer: The patients are from Geneva, but the Principal Investigator and the Sponsor of this study now work in Zurich; or are affiliated to Seattle. As a principle, all authors worked in Geneva some years ago, but are now affiliated elsewhere. In Switzerland, the author affiliations correspond to both, the past and actual universities. We now mention this in one short sentence to avoid misinterpretation (page 2, lines 59-60).
- Skin commensals are here contraposed to “pyogenic pathogens”, instead of using the term “pathogenic bacteria” more often used when Enterobacteriaceae and Pseudomonas aeruginosa are included together with Staphylococcus aureus and streptococci. Please correct line 114 where you can find “pyrogenic” as a mistyping.
Answer: Ok. We now write *pathogenic” instead of “pyogenic” throughout the text.
- If both patient groups with skin commensals and pathogens are equally treated with sensitive antibiotics based on respective antibiograms, how can you expect any difference? A targeted antibiotic therapy is the main reason for not finding any difference in the two groups.
Answer: The theoretical difference in outcome might concern the clinical virulence; and consequently, the final outcomes; even if both groups are treated with targeted regimens. We deny this. However, we now add the good argumentation of Reviewer 1 into the Discussion (page 4, lines 136-138).
- At the end, an oral non-beta-lactam antibiotic therapy is suggested without giving a clear reason since 40% of the skin commensals group was treated with co-amoxiclav. It can be supposed that 34% were treated with co-trimoxazole or doxycycline or rifampicin plus fusidic acid because that was the best choice based on the antibiogram.
Answer: Correct. We emphasize this, as proposed by Reviewer 1 (pages 2-3, lines 90-92).
Reviewer 2 Report
Dear authors, thank you very much for the opportunity to review your paper. I hope after revision it could improve the quality and soundness of the paper.
The research describes a large retrospective cohort of patients admitted with diabetic foot infections (DFIs) in the same hospital. The authors state that all the management were carried out according the IWGDF guidelines. The sample was divided into two different groups: pyogenic bacterias and skin commensals (based on previous literature research). Pyogenic pathogens resulted in worse outcomes than skin commensals, it help clinicians to proper select the therapy of choice depending the culture result.
The research is well described and has scientific soundness, despite this, some changes need to bee made before publication.
-Title and abstract: the paper is a large retrospective case series, authors should name the paper accordingly.
-Based on STROBE statement the paper should follow the following checklist: Abstract-Introduction-Methods (study desing, participants, variables and outcomes and study design)-Results-Discussion-Conclusion. I think authors should adapt the paper to the previous described checklist.
- Describe study design in depth.
- Despite being a retrospective case series, authors should address the Ethics Committee.
-Authors state in methods that wound cultures were taken from foot ulcer bed tissue and pus. Following IDSA and IWGDF wound cultures must be taken from depth samples of the wound after debridement. If authors did not take it accordingly, results may be biassed. I recommend to proper describe/analyze it in methods and if no depth wound cultures were taken, it should be discussed as a limitation.
- Results: I miss descriptive data on bacterial pathogens (at least the most frequent) and additionally, I think authors should add AB therapy and if it was oral or systemic.
- Discussion: authors should address some limitations, for example: study design, sample size calculation, wound culture (not following the IDSA-IWGDF recommendations).
-Conclusions are well described, nonetheless, readers could not reach that message if the paper does not improve methods and results.
Congrats to all the authors, I hope that the revisions could improve the quality of the manuscript.
Author Response
Reviewer 2
Comments and Suggestions for Authors
Dear authors, thank you very much for the opportunity to review your paper. I hope after revision it could improve the quality and soundness of the paper.
Answer: We thank you very much.
The research describes a large retrospective cohort of patients admitted with diabetic foot infections (DFIs) in the same hospital. The authors state that all the management were carried out according the IWGDF guidelines. The sample was divided into two different groups: pyogenic bacteria and skin commensals (based on previous literature research). Pyogenic pathogens resulted in worse outcomes than skin commensals, it helps clinicians to proper select the therapy of choice depending the culture result.
Answer: Yes. This is a correct summary.
The research is well described and has scientific soundness, despite this, some changes need to be made before publication.
Answer: Thank you very much for the helpful suggestions.
-Title and abstract: the paper is a large retrospective case series, authors should name the paper accordingly.
Answer: Ok. It is nevertheless a comparative “study” with more than thousand DFI episodes; and not just a descriptive “case series”. We now mention the retrospective aspect in Title and Abstract (lines 18-19).
-Based on STROBE statement the paper should follow the following checklist: Abstract-Introduction-Methods (study design, participants, variables and outcomes and study design)-Results-Discussion-Conclusion. I think authors should adapt the paper to the previous described checklist.
Answer: Theoretically yes for many journals. However, this journal has its particular order of subchapter, which we have to respect. We followed the author’s instructions.
- Describe study design in depth.
Answer: Agreed. We expand the “study design” by six sentences (page 6, lines 201-210). However, we only introduce the most pertinent information regarding the design.
- Despite being a retrospective case series, authors should address the Ethics Committee.
Answer: We give the authorization number of the Ethical Committee (page 7, lines 256).
-Authors state in methods that wound cultures were taken from foot ulcer bed tissue and pus. Following IDSA and IWGDF wound cultures must be taken from depth samples of the wound after debridement. If authors did not take it accordingly, results may be biased. I recommend to proper describe/analyze it in methods and if no depth wound cultures were taken, it should be discussed as a limitation.
Answer: This is unfortunately a misunderstanding. In our Clinical Pathway (based on the IWGDF guidance and one co-author being the lead of that last IWGDF infection guidance), we only accepted intraoperative, or deep tissue samples during and after the start of debridement. We now emphasize this issue on page 6, lines 211-215.
- Results: I miss descriptive data on bacterial pathogens (at least the most frequent) and additionally, I think authors should add AB therapy and if it was oral or systemic.
Answer: Regarding the microbiology, we already say that “Overall, we detected 68 different microbiological constellations” (page 2, line 71). Regarding the various antibiotic regimens, we already say “45 different regimens, with a median duration of 20 days, of which 5 days were administered parenterally “(page 2, line 83).
Now, we give more detailed microbiological results (page 2, lines 71-74) and equally display graphically the most frequent antibiotic agents used in a new figure (Figure 1).
- Discussion: authors should address some limitations, for example: study design, sample size calculation, wound culture (not following the IDSA-IWGDF recommendations).
Answer: We indeed followed the IWGDF recommendation regarding the intraoperative or deep tissue cultures, because we run a Clinical Pathway for DFI basing on these recommendations (page 6, lines 211-215). For the rest of the points elevated by Reviewer 2, we now add them as “limitations” (page 5, lines 163-175)., although we are not entirely convinced that these are really limitations. For example, our study population involves more than 1000 DFI episodes, by far larger than the majority of other trials in the field of the diabetic foot. So, compared to the literature, the sample size is not a major issue.
-Conclusions are well described, nonetheless, readers could not reach that message if the paper does not improve methods and results.
Answer: We now reword better (pages 9-10, lines 195-203).
Congrats to all the authors, I hope that the revisions could improve the quality of the manuscript.
Answer: Thank you very much.
Reviewer 3 Report
Dear authors
This is an interesting paper about the influence of skin commensals on the clinical outcomes in patients with diabetic foot infections
I think that some minor changes could improve this work.
Results
Among the 392 [39%] episodes of DFO with a 70 positive bone culture, the diagnosis was confirmed by histology in 275 (70%). What type of osteomyelitis was predominant, chronic or acute?
Taking into account the concept of biofilms, it would have been interesting to make a third group with mixed infections (pyogenic bacteria plus skin commensals) and comparing them with the other two groups
From my point of view, a clinical variable that must be taken into account when comparing the groups is the healing time. In other words, do microorganisms present in infections influence healing times? I think this would improve the quality of clinical results.
Methods
Microbiological samples were collected from pus or intraoperative tissue specimens. How were collected the microbiological samples? Curretage, biopsy, swabs?
Limitations
Groups with very different sample sizes (641 versus 54). You should consider this in the limitations of the study.
Discusion
I recommend including this article in the discussion, perhaps it would help explain the role of microorganisms considered as commensals in DFI.
Kallan et al. Strain and species level variation in the microbiome of diabetic wounds is associated with clinical outcomes and therapeutic efficacy
“Cultured wound isolates of S. aureus elicited differential phenotypes in mouse models that corresponded with patient outcomes, while wound “bystanders” such as Corynebacterium striatum and Alcaligenes faecalis typically considered commensals or contaminants also significantly impacted wound severity and healing”
Best regards
Author Response
Reviewer 3
This is an interesting paper about the influence of skin commensals on the clinical outcomes in patients with diabetic foot infections.
Answer: Yes, this is true.
I think that some minor changes could improve this work.
Answer: Thank you very much for your suggestions.
Results
Among the 392 [39%] episodes of DFO with a 70 positive bone culture, the diagnosis was confirmed by histology in 275 (70%). What type of osteomyelitis was predominant, chronic or acute?
Answer: It was “chronic” osteitis. We mention it now on page 2, line 79.
Taking into account the concept of biofilms, it would have been interesting to make a third group with mixed infections (pyogenic bacteria plus skin commensals) and comparing them with the other two groups.
Answer: This would be another aleatory group and beyond the study question of our article. We would like to renounce on it, because it would complicate too much. We evaluate if skin commensals reveal a particular virulence, and outcomes, compared to pyogenic bacteria. We do not investigate the outcomes of mixed infections.
Moreover, in such mixed infections, we ignore which group is clinically predominating. For instance, you might have a mixed infection with Pseudomonas and Corynebacterium. This does not mean that both pathogens contribute equally 50% to the clinical infection. It could be that the Pseudomonas contributes to 90% and that the creation of that group would be artificial; independently of the number of positive microbiological samples for each of them. We would suggest not to mix the retrospective analysis further by mixed groups, and to stick to clearly defined comparator groups.
From my point of view, a clinical variable that must be taken into account when comparing the groups is the healing time. In other words, do microorganisms present in infections influence healing times? I think this would improve the quality of clinical results.
Answer: This is true, albeit we ignore the existence of solid published data that the ulcer healing time specifically is determinant of infection remission. We, however, mention now the lack of the healing time in the Limitations on page 5, lines 163-175. According to experts’ opinion, ulcer healing is influenced by off-loading, compliance and professional wound debridement, and probably only to a minor extent by antibiotic therapy.
Methods
Microbiological samples were collected from pus or intraoperative tissue specimens. How were collected the microbiological samples? Curettage, biopsy, swabs?
Answer: Deep tissue samples, intraoperative tissues and deep pus after the start of debridement. We explain better now (page 6, lines 215-219).
Limitations
Groups with very different sample sizes (641 versus 54). You should consider this in the limitations of the study.
Answer: Agreed. We now mention this methodological limitation on page 5, lines 150-151. However, this disproportion reflects the normal epidemiological nature. Skin commensals are always a minority of the DFI species; in every setting.
Discussion
I recommend including this article in the discussion, perhaps it would help explain the role of microorganisms considered as commensals in DFI. Kallan et al. Strain and species level variation in the microbiome of diabetic wounds is associated with clinical outcomes and therapeutic efficacy.
Answer: We now discuss this new reference (new number 25) on page 5, lines 167-175.
“Cultured wound isolates of S. aureus elicited differential phenotypes in mouse models that corresponded with patient outcomes, while wound “bystanders” such as Corynebacterium striatum and Alcaligenes faecalis typically considered commensals or contaminants also significantly impacted wound severity and healing”
Answer: Yes, this is the above-mentioned article. Reviewer 3 says that skin commensals could also play a role in ulcer healing, which we mention now (page 5, lines 167-175).
Round 2
Reviewer 2 Report
Dear authors,
Congrats for all the work made to review the paper. I hope this effort make an important contribution to our eminent journal.
After a carefully review I realized that authors responded all the questions and amended all required issues.
The paper improved it soundness and I think now the research is ready to be published under the current form.